# UniMo: Unifying 2D Video and 3D Human Motion with an Autoregressive Framework

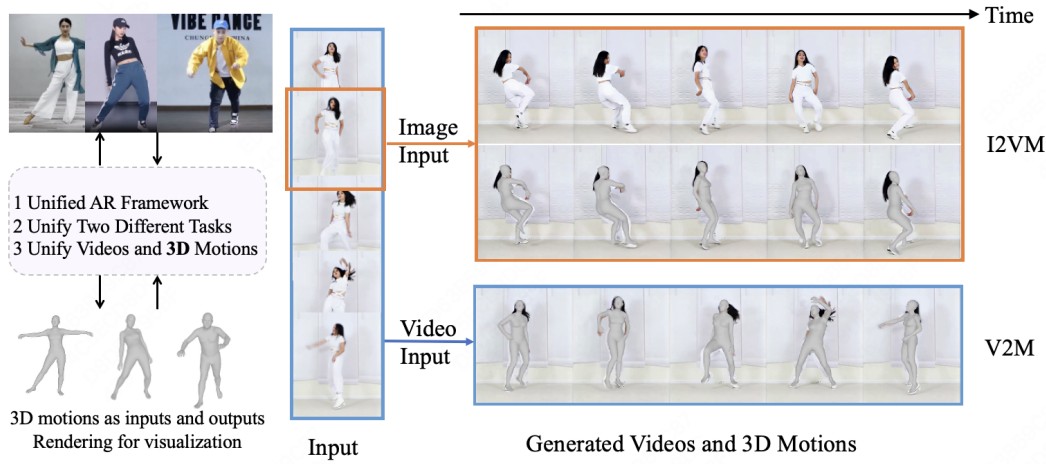

Figure 1: We present UniMo, an innovative autoregressive model for joint modeling 2D human videos and 3D human motions within a unified framework. Left: Unlike existing methods that map 3D motions to 2D maps for video-motion alignment, we directly use 3D motions as inputs and outputs. Right: We unify the I2VM (Image-to-Video-and-Motion) and V2M (Video-to-Motion) tasks within a single transformer framework, demonstrating the effectiveness of the proposed method.

## Abstract

We propose UniMo, an innovative autoregressive model for joint modeling of 2D human videos and 3D human motions within a unified framework, enabling simultaneous generation and understanding of these two modalities for the first time. Current methods predominantly focus on generating one modality given another as the condition or integrating either of them with other modalities such as text and audio. Unifying 2D videos and 3D motions for simultaneous optimization and generation remains largely unexplored, presenting significant challenges due to their substantial structural and distributional differences. Inspired by the LLM's ability to unify different modalities, our method models videos and 3D motions as a unified tokens sequence, utilizing separate embedding layers to mitigate distribution gaps. Additionally, we devise a sequence modeling strategy that integrates two distinct tasks within a single framework, proving the effectiveness of unified modeling. Moreover, to efficiently align with visual tokens and preserve 3D spatial information, we design a novel 3D motion tokenizer with a temporal expansion strategy, using a single VQ-VAE to produce quantized motion tokens. It features multiple expert decoders that handle body shapes, translation, global orientation, and body poses for reliable 3D motion reconstruction. Extensive experiments demonstrate that our method simultaneously generates corresponding videos and motions while performing accurate motion capture. This work taps into the capacity of LLMs to fuse diverse data types, paving the way for integrating human-centric information into existing models and potentially enabling multimodal, controllable joint modeling of humans, objects, and scenes.

# 1 INTRODUCTION

Digital human modeling is a fundamental task in computer vision, where generating consistent human videos and motions is crucial for downstream applications such as virtual reality. Additionally, integrating 3D motions and 2D videos plays a pivotal role in extensive tasks, including human video synthesis and motion capture. For human video synthesis, most methods Shao et al. (2024); Zhu et al. (2024); Lin et al. (2025); Hu et al. (2023) focus on producing human videos that are consistent with input motions. In the realm of motion capture Goel et al. (2023); Shen et al. (2024); Khirodkar et al. (2024), they aim to capture the corresponding 3D motions from input videos. However, the aforementioned methods primarily use one modality as the condition to generate another, without engaging in joint modeling and optimization of both modalities.

This paper focuses on unifying 2D human videos and 3D human motions to achieve joint optimization and generation. Recently, large language models (LLMs) Brown et al. (2020) have been widely applied in various large vision-language models Liu et al. (2023); Bai et al. (2025); Guo et al. (2025); Agarwal et al. (2025), effectively capturing the relationships among different modalities such as audio, text, and vision. Building on this, numerous approaches leverage LLM frameworks to integrate motions with different modalities, including vision Li et al. (2025a); Chen et al. (2024); Li et al. (2025c), text Zhu et al. (2025), and audio Luo et al. (2024). Among these tasks, unifying 2D videos and 3D motions for simultaneous optimization and generation remains unexplored. Inspired by the LLM's ability to unify different modalities, we explore the possibility of transferring this capability to our specific task. This endeavor taps into the capacity of LLMs to fuse diverse data types, paving the way for integrating human-centric information into existing models and potentially enabling multimodal, controllable joint modeling of humans, objects, and scenes.

The primary challenge with 3D motions lies in the lack of explicit spatial correspondence with 2D videos, which inhibits integration via straightforward operations like addition or concatenation Hu et al. (2023). In this paper, we introduce a novel autoregressive (AR) framework for joint modeling of 2D human videos and 3D human motions, achieving simultaneous optimization and generation of these two modalities for the first time. Unlike existing single-task methods, our unified framework efficiently performs both generation and understanding tasks, as illustrated in Fig. 1, further advancing the ability of LLMs to integrate various data types. For the generation task, the model simultaneously produces videos and corresponding 3D motions from a single image. For the understanding task, it captures corresponding 3D motions from video inputs. To unify these two tasks, we design a novel sequence modeling strategy that assigns corresponding tasks based on the input tokens. Moreover, motion tokens differ from visual tokens with respect to token distribution, and the model simultaneously outputs tokens from two modalities, which can easily lead to confusion. Therefore, we design independent embedding layers for each modality to alleviate the distribution gaps, including learnable vocabulary embeddings and positional embeddings.

Another challenge lies in constructing 3D motion representations for seamless integration with visual information within our AR framework. A straightforward approach is to represent 3D motions similarly to visual tokens. Recently, MotionGPT Jiang et al. (2023), SOLAMI Jiang et al. (2025), and Duolando Siyao et al. (2024) have explored 3D motion tokenizers, based on 3D keypoints or SMPL(X) parameters. 3D keypoints are relatively simple but insufficient to represent complex human motions. Therefore, we use SMPL-X Pavlakos et al. (2019) to model human motions, incorporating 3D parameters like body shapes, translation, global orientation, and body poses. However, applying current tokenizers directly to our task introduces two main challenges. Firstly, most methods employ temporal compression to reduce resource usage, which is effective for motion-text alignment but results in a token quantity imbalance in video-motion modeling. Secondly, many approaches segment the human body into multiple parts, processing each separately with several VQ-VAEs. Although this improves reconstruction accuracy, it results in multiple sets of motion tokens, thus increasing complexity for our unified model. To overcome these challenges, we propose a novel 3D motion tokenizer that models all SMPL-X parameters using a single VQ-VAE, complemented by a novel temporal expansion strategy to enhance reconstruction accuracy and balance the quantity of vision tokens. The proposed tokenizer generates motion tokens while ensuring accurate reconstruction, laying the foundation for effective multimodal fusion through AR.

By leveraging the proposed motion tokenizer and AR model, our approach consistently generates both videos and motions across two tasks, illustrating the potential of modeling 3D motions and

2D videos within a unified AR framework. This work not only explores the capacity of LLMs to fuse diverse data types but also establishes a foundation for embedding human-centric information into existing architectures, potentially enabling multimodal, controllable joint modeling of humans, objects, and scenes. We summarize our contributions as follows,

- A novel LLM-based framework jointly models 3D human motions and 2D human videos, enabling simultaneous generation and optimization of both modalities for the first time.

- A novel 3D motion tokenizer employs a temporal expansion strategy to effectively quantize and reconstruct SMPL-X parameters, laying the foundation for modal integration and motion generation.

- A unified autoregressive model, featuring a novel sequence modeling and independent embedding strategy, integrates two distinct tasks within a single transformer model, effectively alleviating distribution gaps and merging both modalities.

- Demonstration of the simultaneous generation and understanding of both videos and motions.

## 2 RELATED WORKS

**Human Video Synthesis.** The objective is to generate a corresponding video given a single human image and a sequence of driving motion. Current approaches predominantly utilize diffusion-based methods to tackle this task, often employing UNet Hu et al. (2023); Zhu et al. (2024); Xu et al. (2024); Wang et al. (2024a); Chang et al. (2025); Kim et al. (2024); Zhang et al. (2024) or DiT Shao et al. (2024); Lin et al. (2025); Shao et al. (2025); Ding et al. (2025) structures. To incorporate the driving motions, they typically utilize 2D maps such as skeleton maps, normal maps, and densepose maps Karras et al. (2023). Although these methods can produce vivid and consistent human videos, they only align 2D motions with visual latents at the input stage and lack optimization of 3D motions.

**Human Motion Capture.** Motion capture Shen et al. (2024); Goel et al. (2023); Kanazawa et al. (2018); Rajasegaran et al. (2022); Kanazawa et al. (2019); Kocabas et al. (2020); Luo et al. (2020); Khirodkar et al. (2024) is a classic task aiming at extracting corresponding human motions from video inputs. For instance, HMR Kanazawa et al. (2018) utilizes a CNN to regress SMPL parameters, while 4DHumans Goel et al. (2023) introduces a fully transformer-based model based on an enhanced HMR and 3D tracking system. Besides, GVHMR Shen et al. (2024) estimates human poses in a novel gravity-view coordinate to reduce ambiguity in image-pose mapping. Typically, they commence video preprocessing by tracking humans, detecting keypoints, and extracting features, followed by regressing motion parameters from these features. Besides, they primarily focus on the transfer from videos to motions, lacking emphasis on the joint modeling of the two modalities.

**Human Video-Motion Joint Tasks.** Some methods Li et al. (2025a); Chen et al. (2024); Li et al. (2025c) employ joint modeling of videos and motions to enhance the understanding of human behavior. However, these approaches merely integrate the two modalities into a unified representation at the input stage, lacking the capability to simultaneously generate both. VideoJAM Chefer et al. (2025), AnimaX Huang et al. (2025), and OmniVDiff Xi et al. (2025) introduce an appearance-motion aligning framework based on diffusion models, which represent motion information using 2D motion maps. While 2D motion maps are flexible and can be easily represented as RGB videos, they inevitably suffer from the loss of crucial 3D spatial information. SViMo Dang et al. (2025) combines visual priors with dynamic constraints to simultaneously generate hand-object videos and motions. However, they adopt diffusion frameworks and focus on the point cloud generation.

**Human 3D Motion Tokenizer.** The human motion tokenizer Ding et al. (2025) is designed to compress and convert raw motion data, such as keypoints and SMPL-X parameters, into motion tokens. MotionGPT Jiang et al. (2023) and SOLAMI Jiang et al. (2025) pre-train the 3D human motion tokenizers using the VQ-VAE architecture. However, their tokenizers require complex and extensive data processing, including orientation adjustments, foot contact modifications, and the application of forward/inverse kinematics, which demands predefined human kinematic chains tailored to specific datasets. Recently, Duolando Siyao et al. (2024) introduces a simplified motion tokenizer that uses raw 3D joint coordinates as inputs. Nevertheless, they independently process body poses and translation, and 3D joints remain insufficient to fully capture complex human motions.

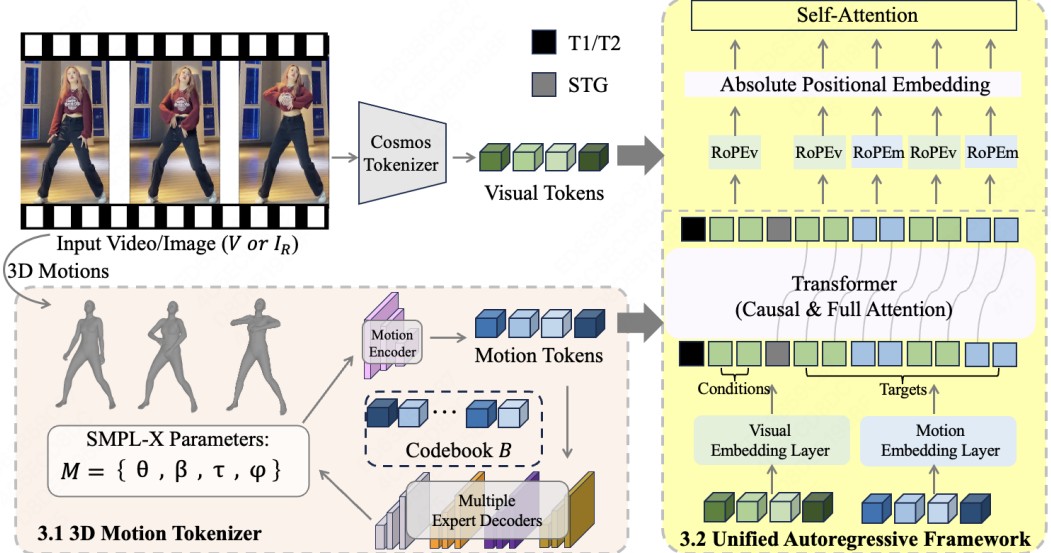

Figure 2: Overview of UniMo. Left: We introduce a 3D motion tokenizer that is responsible for quantizing raw 3D motions $M$ into motion tokens corresponding to visual tokens and accurately reconstructing back to $M$ from these tokens. The 3D motion tokenizer comprises a motion encoder, a learnable codebook $B$, and multiple expert decoders. Right: Given visual tokens and motion tokens, we propose an AR transformer framework with new sequence modeling strategies to unify the two modalities, enabling the execution of two distinct tasks (only the I2VM task is illustrated in the figure). To better integrate the two types of tokens, we propose independent vocabulary embedding layers and positional embedding layers.

**LLM-based Video and Motion Models.** Recently, autoregressive models Wu et al. (2025); Wang et al. (2024b); Chen et al. (2025); Wu et al. (2024) based on transformer architectures have demonstrated impressive results in multimodal modeling. Cosmos Agarwal et al. (2025) approaches world simulation generation as a next-token prediction task, akin to language modeling, and incorporates text embeddings using cross-attention. Additionally, many methods Jiang et al. (2023; 2025); Chen et al. (2024); Li et al. (2025b) leverage LLMs to achieve unified generation and understanding of motions alongside multiple modalities such as text and audio.

## 3 METHOD

The overall pipeline of UniMo is illustrated in Fig. 2. Our goal is to model 3D motions and 2D videos without relying on 2D motion maps, achieving simultaneous generation and optimization of them. Specifically, UniMo employs a unified AR framework to integrate the two modalities and implement two distinct tasks for validating the generation and understanding capabilities. In the image-to-video-and-motion task (I2VM), given a single reference image $I_R$, the goal is to generate subsequent $T$-frame videos along with corresponding 3D motions $M_{k=1}^T$. In the video-to-motion task (V2M), given a video sequence $V_{k=1}^T$, the objective is to capture the corresponding 3D motions.

We first introduce a 3D motion tokenizer tailored for our task in Sec. 3.1, which is responsible for quantizing raw 3D motions $M$ into motion tokens corresponding to visual tokens. Additionally, this tokenizer can accurately reconstruct $M$ from the quantized tokens, establishing a foundation for joint modeling. Then, we detail how autoregressive modeling can unify visual and motion tokens within a transformer architecture in Sec. 3.2, encompassing task-specific sequence modeling strategies and independent embeddings adapted to various modalities. By simultaneously optimizing two modalities and tasks, the model enhances their correspondence and improves generative and understanding capabilities. After that, we discuss the training and inference strategies utilized for multi-task unification in Sec. 3.3.

## 3.1 3D Motion Tokenizer

To integrate 3D motions $M$ with visual tokens and enable their generation, it is necessary to develop a tokenizer that accepts $M$ as input, quantifying them into discrete tokens akin to visual tokens, while also being capable of reconstructing the original $M$. MotionGPT Jiang et al. (2023), SOLAMI Jiang et al. (2025), and Duolando Siyao et al. (2024) have explored 3D motion tokenizers for this target, based on 3D keypoints or SMPL(X) parameters. 3D keypoints are relatively simple but insufficient to fully represent complex 3D human motion information. Moreover, obtaining SMPL(X) parameters offers greater value in motion generation tasks. MotionGPT and SOLAMI are SMPL(X)-based methods that require complex processing of motions, including orientation adjustments, foot contact modifications, and the application of forward/inverse kinematics, making it difficult to generalize across diverse datasets. Additionally, SOLAMI divides the human body into multiple parts and designs several VQVAEs to independently learn different segments and translations. This configuration generates multiple sets of tokens, increasing complexity in our unified model.

To address this, as shown in the left box in Fig. 2, we use SMPL-X models Pavlakos et al. (2019) to comprehensively represent 3D human motions and propose a novel 3D motion tokenizer tailored to our task. Specifically, SMPL-X is built to represent human motions through parameterized body poses ($\theta \in \mathbb{R}^{T \times 63}$), shape coefficients ($\beta \in \mathbb{R}^{T \times 10}$), global orientation ($\phi \in \mathbb{R}^{T \times 3}$), and translation ($\tau \in \mathbb{R}^{T \times 3}$). The proposed tokenizer is capable of taking the entire set of $M = (\theta, \beta, \phi, \tau)$ as inputs, quantizing them into discrete tokens, and accurately reconstructing the SMPL-X parameters from motion tokens. Inspired by Duolando Siyao et al. (2024), we employ a VQ-VAE structure comprising an encoder, a learnable codebook, and multiple expert decoders. We first cascade $M$ along the last dimension channel, then the encoder uses 1D convolutions transforming them into high-dimensional semantic features $F \in \mathbb{R}^{T' \times C}$, where $T' = T/s$ and $C$ is the number of channels. In data processing, the absolute position of the first frame is preserved, while subsequent frames are transformed into velocity representations by subtracting the position of the preceding frame, thereby reinforcing temporal continuity. During reconstruction, the original positions can be restored using prefix sum techniques. Subsequently, the sequence $F_{k=1}^{T'} = \{f_1, f_2, ..., f_{T'}\}$ is quantized by replacing each $f_k$ with the nearest element in the codebook $B$, transforming it into a discrete sequence of tokens. Finally, we employ four expert decoders consisting of 1D convolutions to individually reconstruct parameters $\theta$, $\beta$, $\tau$, and $\phi$ from tokens.

Notably, most methods employ temporal compression to reduce resource usage, which is effective for motion-text alignment but results in a token quantity imbalance in our task. Besides, SMPL-X parameters are more complex than 3D keypoints, resulting in greater learning difficulty and poorer performance on motion metrics compared to keypoints Jiang et al. (2025). To address this issue, we adopt an expansion strategy for temporal processing. By setting $s = 1/36$, we represent the SMPL-X parameters for one frame with 36 discrete tokens. This serves three purposes: (1) Visual tokens significantly outnumber motion tokens, thus expanding motion tokens helps balance the disparity in their quantities to some extent; (2) One of our objectives is the accurate reconstruction of SMPL-X parameters, and expanding tokens can improve accuracy; (3) With a relatively small number of parameters for the motion tokenizer, expanding the tokens only imposes a small burden (about 30M).

## 3.2 Unified Autoregressive Framework

**Unified Motion-Visual Representation.** Building on the impressive performance of Cosmos Agarwal et al. (2025) in autoregressive (AR) video modeling, we adopt the Cosmos AR framework as our backbone model. We utilize the Cosmos tokenizer to quantize and compress the videos into visual tokens with a compression rate of 8x16x16. However, in Cosmos, multimodal sequence modeling is accomplished through cross-attention, which prevents simultaneous generation and optimization of another modality. Inspired by LLMs Xie et al. (2025); Wang et al. (2024b), we structure 3D motion tokens and visual tokens into a unified sequence following an interleaved motion-video format. Given that our framework involves I2VM and V2M tasks, we introduce special tokens to identify different tasks. For V2M task, we format the sequence as:

$$[T1] \quad [Vt_1 \quad Vt_2 \quad ... \quad Vt_N] \quad [STG] \quad [Mt_1 \quad Mt_2 \quad ... \quad Mt_M] \tag{1}$$

where $T1$ means V2M task. $Vt$ and $Mt$ represent the visual and motion tokens, respectively. $STG$ marks the beginning of generation, with the conditional sequence placed before it and the target se-

Figure 3: Comparison with the baseline on I2VM tasks. We present results in temporal order from left to right, sampling one frame every 10 generated frames. In our results, the simultaneously generated 3D motions are rendered and visualized in the top-left corner, while the baseline model lacks motion generation capabilities. Given a single image as input, the baseline tends to generate results with minimal motion amplitude, often bordering on stillness.

quence positioned after it. $N$ and $M$ represent the number of visual and motion tokens, respectively. For I2VM task, we format the sequence as:

$$[T2] \quad [It] \quad [STG] \quad [Vt_1] \quad [Mt_1] \quad [Vt_2] \quad [Mt_2] \quad ... \quad [Vt_N] \quad [Mt_M] \tag{2}$$

where $T2$ means I2VM task. $It$ is the single reference image tokens. The sequence formats above are flexible by using a task-specific token at the beginning to distinguish two tasks, and employing the $STG$ special token to separate the conditions from the targets within different tasks. Notably, in Eq. 2, we define the target sequence as interleaved visual tokens and motion tokens. This design strengthens the model's capability to integrate both modalities simultaneously, enabling the generation process to leverage all previously incorporated modalities.

**Vocabulary Embedding Layers.** In the Cosmos AR model, sequence modeling involves only visual modality, and thus uses a single embedding layer to process all discrete tokens. However, due to the inherent gaps between motion and visual tokens, using a single embedding layer may lead to distribution entanglement. Additionally, simultaneous generation of two modalities within one framework raises the possibility of output modality confusion. To address these issues, as shown in the right box in Fig. 2, we employ two separated learnable embedding layers, one for visual tokens and the other for motion tokens.

**Positional Embedding Layers.** The Cosmos AR model employs two complementary positional embedding mechanisms to convey spatial and temporal information across the network: 3D factorized absolute positional embedding (APE) captures absolute coordinates, while 3D factorized Rotary Position Embedding (RoPE) addresses relative positions. In our work, we retain APE to simultaneously establish the absolute positional relationship between two modalities. For RoPE, similar to the separated vocabulary embedding layers, we implement two distinct RoPE mechanisms to independently process each modality, thereby establishing positional relationships within each modality at different positions (right box in Fig. 2). Specifically, for queries ($Q$) and keys ($K$) in attention operations, we select the corresponding RoPE based on the modality of the current position:

$$\hat{Q} = (RoPE_m(Q_m) \oplus RoPE_v(Q_v)) + APE(Q_e) \tag{3}$$

$$\hat{K} = (RoPE_m(K_m) \oplus RoPE_v(K_v)) + APE(K_e) \tag{4}$$

where $Q_e$ represents the entire sequence, $\hat{Q}$ represents the results after applying positional embedding. $RoPE_m$ and $RoPE_v$ represent motion RoPE and visual RoPE respectively. $Q_m$ and $Q_v$ represent motion tokens and visual tokens respectively, and $\oplus$ represents concatenate operations. ($K$ likewise)

### 3.3 TRAINING AND INFERENCE STRATEGY

**Training.** UniMo is trained with a two-stage training approach. In the first stage, we train the 3D motion tokenizer in an end-to-end way using 3D motion data, and the training loss is:

$$\mathcal{L}_{VAE} = \mathcal{L}_{rec}(M', M_{gt}) + \lambda \|F - sg(B)\| + \|sg(F) - B\| \tag{5}$$

Table 1: Evaluation of the 3D motion tokenizer on Human4DiT-Video dataset.

| Methods | MPJPE ↓ | PA-MPJPE↓ | PVE↓ | Accel↓ |
|---------|---------|-----------|------|--------|
| SOLAMI | 24.3354 | 14.7212 | 29.6462 | 7.7384 |
| Our | **8.6344** | **5.3876** | **10.7010** | **2.4632** |

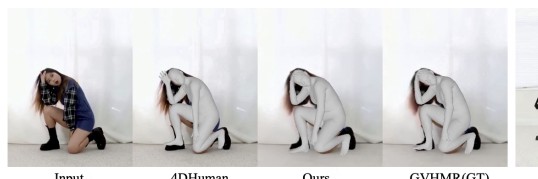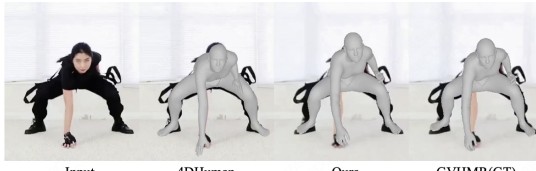

Figure 4: Comparison with different methods on the V2M task. It demonstrates that our approach achieves results comparable to current state-of-the-art methods.

where $\mathcal{L}_{rec}$ is $l_1$ loss between the predicted value $M'$ and the real value $M_{gt}$. sg means stop gradient operation and $\lambda$ is the trade-off parameter. Similar to Duolando, we add velocity and acceleration to perform $\mathcal{L}_{rec}$. Notably, the motion encoder is used only during the training phase.

In the second stage, the parameters of tokenizer are frozen to serve as a quantizer, supplying discrete motion tokens for training the AR model. Attention masks are crucial for AR models training, with most methods utilizing causal masks to ensure that the current token can only attend to preceding tokens. In our task, we apply causal masks to the target sequence, while employing full masks on the conditional sequence to enhance bidirectional context awareness. In addition, we combine the data from the two tasks in equal proportions to unify the multimodal and multi-task, and train the AR model in an end-to-end way. The loss is the cross-entropy:

$$\mathcal{L}_{AR} = -\sum_{i=1}^{L} \log p(q_i|q_{<i}, c) \tag{6}$$

where $L$ is target sequence length. $q_i$ is the i-th token in the sequence and $c$ means the conditions.

**Inference.** During the inference phase, we employ the AR model, the diffusion decoder from Cosmos, the continuous tokenizer decoder in Cosmos, and the multiple expert decoders from our 3D motion tokenizer. For both tasks, we use the conditional sequence as input, first filling the KV cache during the prefill phase and subsequently utilizing the cached representations for next-token prediction. The diffusion decoder enhances visual quality by converting discrete representations into continuous ones within the diffusion latent space. Continuous representations are then fed into the continuous tokenizer decoder to generate the final video outputs. Additionally, the multiple expert decoders reconstruct the generated motion tokens into SMPL-X parameters, ensuring accurate motion representation.

## 4 EXPERIMENT

### 4.1 SETTINGS

**Metrics.** We employ distinct metrics to evaluate the generated videos and motions. For video evaluation, similar to VideoJAM Chefer et al. (2025), we utilize VBench Zheng et al. (2025) to analyze various disentangled features, including appearance and visual motion attributes. Please refer to our supplementary paper for more results. We evaluate the generated 3D motions in two ways. For I2VM tasks, we use FID and diversity (DIV) metrics to compare the distribution of generated motions with ground-truth motions. In line with Duolando Siyao et al. (2024), we derive FID and diversity from motion features in AIST++ Li et al. (2021). For V2M tasks, following WHAM Shin et al. (2024), we report metrics such as MPJPE, PA-MPJPE, PVE, and Accel.

We train our model on Human4DiT-Video Shao et al. (2024), a dataset comprising 10k in-the-wild monocular video clips with corresponding motion sequences. Besides, we observe temporal jitter in the motion dataset within Human4DiT-Video, prompting us to use GVHMR Shen et al. (2024) to

Table 2: Comparison of I2VM tasks. For motion-video consistency, GVHMR is used to extract motions as pseudo ground truth. Motion diversity is assessed using the metrics from Duolando. The generated videos are evaluated with VBench, focusing on appearance (App.) and visual motion (Mot.). The baseline (Cosmos) lacks the ability to generate motion and doesn't involve consistency.

| Method | Video-Motion Consistency | | | | Motion Diversity | | Video Quality | |
|---|---|---|---|---|---|---|---|---|
| | MPJPE $\downarrow$ | PA-MPJPE $\downarrow$ | PVE $\downarrow$ | Accel $\downarrow$ | FID $\downarrow$ | DIV $\uparrow$ | App. $\uparrow$ | Mot. $\uparrow$ |
| Baseline | - | - | - | - | 57.3624 | 5.7160 | 0.8071 | 0.8126 |
| Ours | 41.3058 | 30.9548 | 47.3111 | 3.9984 | **27.3984** | **12.2522** | **0.8516** | **0.9441** |

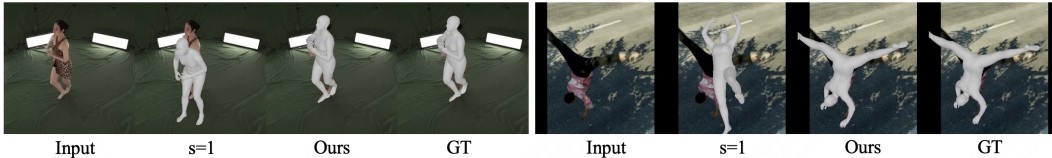

Input      s=1      Ours      GT      Input      s=1      Ours      GT

Figure 5: Ablation experiments of the number of tokens in 3D motion tokenizer. It is difficult for s=1 to represent the body shapes, body poses, global orientation, and translation simultaneously.

re-extract the 3D motion data. For evaluation, we select 300 single-human clips from Human4DiT-Video Shao et al. (2024), DNA-Rendering Cheng et al. (2023), 3DPW von Marcard et al. (2018), RICH Huang et al. (2022), and BEDLAM Black et al. (2023) as the testset. It is important to note that our model actually uses 3D motions (SMPL-X) as both inputs and outputs. For intuitive visualization, we render the generated 3D motions into 2D maps in all visual results. In addition, due to the poor visual effect of video tokenizer in Cosmos, we use SeedVR Wang et al. (2025) to enhance the visual quality of generated videos, which does not disrupt our core goal of joint modeling the two modalities. Please refer to our supplementary paper for more implementation details.

## 4.2 EVALUATIONS AND COMPARISONS

**3D Motion Tokenizer.** Unlike existing methods such as SOLAMI, which processes the human body into parts with multiple VQ-VAEs, we utilize a single VQ-VAE to process the entire SMPL-X parameters. To ensure the reconstruction accuracy, we propose a temporal expansion strategy. Specifically, we compare our approach with SOLAMI after fine-tuning on the Human4DiT-Video dataset. As shown in Tab. 1, our tokenizer exhibits superior performance in reconstruction accuracy. For additional visual results, please refer to the supplementary paper.

**Image-to-Video-Motion Task.** To verify the effectiveness of the proposed method, we conduct comparisons with the baseline Cosmos AR model Agarwal et al. (2025). As illustrated in Fig. 3, when provided with a single image input, the baseline tends to generate results with minimal motion amplitude, often approaching stillness, whereas our method yields dynamic results. From the figure, our method generates results with concurrent changes in translation and body poses. In the quantitative comparisons, we use VBench Zheng et al. (2025) to evaluate the generated videos, focusing on appearance (App.) and visual motion (Mot.) aspects. Besides, the generated motions are evaluated from two dimensions: consistency with the generated videos and motion diversity. For motion-video consistency, we use GVHMR to extract motions from the generated videos as pseudo ground truth. As the baseline model, Cosmos lacks motion generation capabilities, preventing us from evaluating consistency. For motion diversity, we extract motions from generated videos from Cosmos using GVHMR to represent its motion outputs. The quantitative results, shown in Tab. 2, are consistent with the visual results. Our predicted motions align well with the generated video, and are more realistic and diverse. Additionally, our generated videos outperform the baseline. Detailed video quality metrics are provided in the supplementary material.

**Video-to-Motion Task.** To demonstrate the potential of joint modeling, we validate the effectiveness of the V2M task across three datasets: 3DPW, RICH, and Human4DiT-Video. Notably, as our primary focus is on exploring the possibility of joint modeling 3D motion and visual information, our experiments are conducted on single-person cases. As illustrated in Fig. 4 and Tab. 3, our method achieves results comparable to current state-of-the-art methods. For more quantitative comparisons and results, please refer to the supplementary paper.

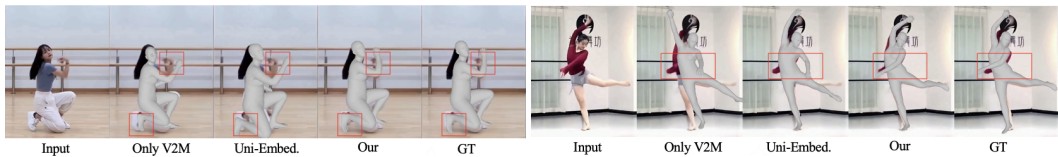

Figure 6: The ablation experiment on the V2M task. We compare our method against single-task training (Only V2M) and the approach without independent embedding (Uni-Embed.).

Table 3: Evaluation of the V2M task on Human4DiT-Video dataset. Since we use SMPL-X extracted by GVHMR as pseudo ground truth, GVHMR is excluded from metric calculation.

| Methods | MPJPE↓ | PA-MPJPE↓ | PVE↓ | Accel↓ |
|---------|--------|-----------|------|--------|
| GVHMR | - | - | - | - |
| 4DHuman | 56.0701 | 35.4769 | 67.7157 | 15.7393 |
| Our | **43.2689** | **28.1528** | **52.0143** | **4.5647** |

## 4.3 ABLATION STUDIES

**3D Motion Tokenizer.** To improve reconstruction accuracy and balance the number of video tokens, we propose expanding the motion token representation by using 36 tokens for each frame's SMPL-X parameters $M$. Specifically, our experiments utilize GVHMR to extract the motions as pseudo ground truth. As shown in Fig. 5, we compare our configuration with the one where one token corresponds to a single frame ($s = 1$). Evidently, $s = 1$ is inadequate for modeling all SMPL-X parameters simultaneously, resulting in errors in translation, global orientation, and body pose representation. Through quantitative and qualitative comparisons, we demonstrate that our 3D motion tokenizer effectively quantizes and reconstructs $M$, thereby establishing a solid foundation for AR training. We provide more detailed quantitative comparisons and analyses in the supplementary paper.

**Independent Embeddings.** We employ independent embeddings for discrete tokens and positions to mitigate distribution disparities between the two modalities, thereby avoiding confusion when simultaneously outputting video and motion tokens within the same transformer model. As illustrated in Fig. 6, we attempt to utilize a single embedding layer for both visual and motion tokens, alongside a unified RoPE embedding for the entire sequence. It is evident that for complex motions, unified embedding is less effective in capturing local details, such as those involved in squats. Please refer to our supplementary paper for more qualitative and quantitative results.

**Single Task.** We integrate two tasks, I2VM and V2M, within the same transformer framework to achieve two objectives. First, the two tasks are used to validate the effectiveness of the proposed method. Second, we observe that training two tasks together yields better results than training each one individually, indicating a synergistic effect between the two tasks. As illustrated in Fig. 6, independent training of the V2M task fails to attain enhanced precision. Please refer to our supplementary paper for more qualitative and quantitative results, including the I2VM task.

## 5 CONCLUSION

In this work, we introduce an innovative autoregressive model for the joint modeling of 2D human videos and 3D human motions within a unified framework. We propose a novel 3D motion tokenizer to establish a direct connection between 3D motions and visual information, thereby avoiding the use of 2D motion maps as the proxy. By designing task-specific sequence modeling strategies and independent embedding methods, the proposed approach effectively integrates the two modalities and tasks. Our extensive experiments show that the model generates corresponding videos and motions while capturing accurate motions. In addition, both quantitative and qualitative comparisons demonstrate that our method achieves performance comparable to state-of-the-art methods. This work taps into the capacity of LLMs to fuse diverse data types, paving the way for integrating human-centric information into existing models and potentially enabling multimodal, controllable joint modeling of humans, objects, and scenes.

**Ethics Statement.** Our work focus on generating human videos and motions purely from a technical standpoint, with no intent for malicious use. Nevertheless, we acknowledge the potential for misuse, such as the creation of fake videos. To mitigate this, it is imperative that synthetic motions and videos are clearly labeled to reflect their artificial origin.

**Reproducibility Statement.** The datasets we used, which include Human4DiT-Video, 3DPW, and RICH, as well as the network baselines, specifically Duolando for tokenizer and Cosmos for AR, are publicly available. Comprehensive descriptions of dataset processing, network structure improvement, and training parameters are provided in the methods section and supplementary paper.

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

## A   MORE DETAILS

**Implementation Details.** We use Cosmos-AR-4B Agarwal et al. (2025) model as the backbone, and train our model on Human4DiT-Video Shao et al. (2024), a dataset comprising 10k in-the-wild monocular video clips with corresponding motion sequences. Notably, we observe temporal jitter in the motion dataset within Human4DiT-Video, prompting us to use GVHMR Shen et al. (2024) to re-extract the 3D motion data. Additionally, all videos are processed to a resolution of 512x512 using pad-resize operations. For the 3D motion tokenizer, the model is trained on data segmented into clips of 32 frames each, utilizing a single NVIDIA A100 GPU with the AdamW optimizer at a learning rate of 4e-5 and a batch size of 128. For AR framework, the data is divided into clips of 65 frames serving as inputs. The AR training utilizes 8 NVIDIA A100 GPUs, each employing the AdamW optimizer with a learning rate of 1e-4 and a batch size of 1 per GPU.

**Network details.** The 3D motion tokenizer is built using 1D convolutional layers. The encoder is composed of four blocks, each featuring a transposed convolution and a ResNet block with consistent channel dimensions. The transposed convolution is responsible for expanding the temporal dimension, while the ResNet blocks enhance the network's depth. We find that a codebook size of 512 effectively balances high utilization and reconstruction accuracy. Each multi-expert decoder also contains four blocks, consisting of a convolution layer and a ResNet block with unchanged channels. The convolution restores the temporal sequence, and the ResNet blocks provide additional depth to the network. Input tensors $M_a$ are structured with the temporal dimension placed last, formatted as $bs \times C \times T$. After processing through the encoder, $M_a$ are transformed into features of size $bs \times 512 \times (T \times 36)$ and subsequently quantized into $bs \times (T \times 36)$ discrete motion tokens. The decoders then reconstruct these tokens back to $bs \times C \times T$ where $C$ signifies specific channels, such as $C = 3$ for the translation decoder.

We build AR model based on Cosmos-AR-4B Agarwal et al. (2025), and the configuration is shown in Tab 4.

**Inference Performance.** During inference, to increase the diversity of the results, we set the top-p parameter to 0.8 and the temperature to 1. In diffusion decoder, we use 15 DDIM steps Song et al. (2020) with a classifier-free guidance (CFG) scale of 2. For the V2M task, capturing a 3-second video consisting of 65 frames takes approximately 30 seconds. In the I2VM task, generating a 3-second video with corresponding motions at a resolution of 512x512 takes around 100 seconds.

Table 4: The configuration of AR model.

| | |
|---|---|
| Number of Transformer Layers | 16 |
| Number of Tokens | 11558 |
| Model Dimension | 4096 |
| Vocabulary Size | 64000 |
| FFN Hidden Dimension | 14336 |
| Number of Attention Heads | 32 |
| Base Learning Rate | 1e-4 |
| Number of Key / Value Heads | 8 |

**VBench Metrics.** Inspired by VideoJAM Chefer et al. (2025), we employ VBench Zheng et al. (2025) to evaluate the quality of generated videos across two dimensions: appearance and visual motion. Appearance metrics include subject consistency, background consistency, and aesthetic quality. Visual motion metrics comprise motion smoothness, temporal flickering, and dynamic degree. It is important to note that visual motion pertains to motion exhibited in 2D RGB videos, distinct from the 3D motions we generate.

Subject consistency is crucial for evaluating whether human identity remains consistent in the generated video, especially amid complex motion dynamics. Background consistency assesses whether non-human content remains stable as the human subject moves, also indicating the AR model's effectiveness in focusing on human regions while maintaining stability in surrounding content. Aesthetic quality measures the overall artistic appeal and perceived beauty of the video.

Motion smoothness evaluates inter-frame continuity in the generated videos. Temporal flickering examines the consistency of temporal sequences by considering local and high-frequency details. Dynamic degree is essential for gauging the amplitude of human movements in the video, preventing the production of static videos that could result in high scores in motion smoothness and temporal flickering.

## B  MORE ABLATIONS

**3D Motion Tokenizer.** We propose a novel 3D motion tokenizer that takes $M = (\theta, \beta, \phi, \tau)$ as input for quantization and accurate reconstruction. In our experiments, we configure the codebook size $B$ to 512 and set the temporal compression rate $s$ to 1/36, meaning that each frame of SMPL-X parameters is represented by 36 tokens. We conduct additional experiments to thoroughly explore the effects of varying $B$ and $s$. As illustrated in Tab. 5, increasing the codebook size results in decreased utilization, accompanied by a decline in reconstruction accuracy. Additionally, when $s$ is large (such as 1/1), the representation becomes inadequate for capturing complex motions. Besides, we evaluate the model parameters for both configurations and find that increasing the number of tokens does not lead to a significant computational burden. Through quantitative and qualitative comparisons, we demonstrate that our 3D motion tokenizer effectively quantizes and reconstructs $M$, thereby establishing a solid foundation for AR training.

Table 5: Evaluation of our 3D motion tokenizer on different network settings. For codebook utilization, we randomly select 100 samples from the testset and calculate the proportion of different tokens, represented as $n/B$, where $n$ represents the number of different tokens.

| Methods | MPJPE↓ | PA-MPJPE↓ | PVE↓ | Accel↓ | Model Param. | Codebook Util. |
|---|---|---|---|---|---|---|
| s=1/1 B=512 | 136.6638 | 81.7294 | 155.1556 | 6.4386 | 90.49 M | 80.08% |
| s=1/8 B=512 | 69.1781 | 52.2387 | 97.2101 | 5.3642 | 95.31 M | 91.99% |
| s=1/24 B=256 | 52.9926 | 45.4241 | 85.0266 | 4.5206 | 86.75 M | 97.66% |
| s=1/24 B=512 | 10.4653 | 6.7306 | 12.7147 | 2.7336 | 101.63 M | 96.09% |
| s=1/24 B=1024 | 47.4062 | 28.9208 | 53.6114 | 4.5009 | 121.51 M | 78.52% |
| s=1/36 B=256 | 26.8738 | 13.3755 | 35.1941 | 3.1909 | 108.25 M | 99.61% |
| s=1/36 B=512 | **8.6344** | **5.3876** | **10.7010** | **2.4632** | 123.13 M | 98.83% |
| s=1/36 B=1024 | 30.8471 | 23.0493 | 44.2118 | 3.7807 | 138.01 M | 86.13% |
| s=1/54 B=512 | 11.8932 | 7.7327 | 17.878 | 2.9688 | 155.71 M | 99.80% |

When modeling SMPL-X parameters, selecting an appropriate rotation representation is crucial to ensuring stability and efficiency. The rotation matrix offers a complete and algebraic representation but suffers from constraints like orthogonality and consumes more memory with its nine parameters, potentially leading to numerical instabilities. On the other hand, the axis-angle representation intuitively combines a rotation axis with an angle, providing a more compact form with only three parameters. However, challenges arise with axis-angle when handling rotations near zero angles or interpolating accurately across rotations. To address these limitations, the 6D representation stands out as a balanced choice. It avoids the orthogonality requirements of rotation matrices, allowing smoother interpolation and maintaining numerical stability. Although less intuitive, the 6D representation effectively supports learning-based approaches. After evaluating these methodologies, the 6D representation is selected for its compactness and robustness—qualities essential to the complexity of SMPL-X parameter modeling.

Table 6: Evaluation of our 3D motion tokenizer on different SMPL-X inputs.

| Methods | MPJPE↓ | PA-MPJPE↓ | PVE↓ | Accel↓ |
|---|---|---|---|---|
| Rotation Matrix | 12.3837 | 6.2096 | 11.4083 | 2.5396 |
| Axis-Angle | 13.4334 | 6.0989 | 16.2724 | 3.0279 |
| 6D Representation | **8.6344** | **5.3876** | **10.7010** | **2.4632** |

As shown in Tab. 6, although the differences among the three representations are minimal, the 6D representation presents superior reconstruction performance overall.

**Effectiveness of Multi-Task Training.** We integrate the I2VM and V2M tasks into mixed training within a single transformer model, enabling simultaneous execution of both tasks. This approach serves two key purposes: first, it demonstrates the effectiveness of integrating 3D motions and 2D videos; second, multi-task learning enhances training stability and convergence. As illustrated in Fig. 7, multi-task learning results in lower loss compared to single-task training. Furthermore, Fig. 8 and Fig. 9 show that multi-task learning also achieves superior visual performance. Tab. 8 and Tab. 7 align with the visual results, demonstrating the effectiveness of multi-task training.

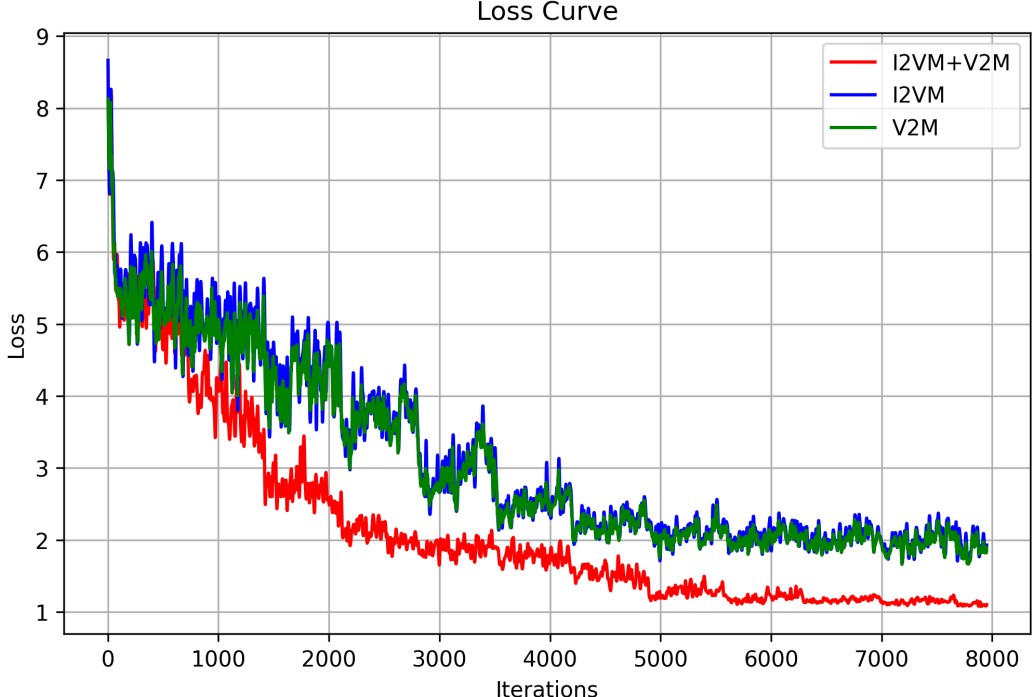

Figure 7: Comparison of losses between single-task and multi-task training.

Table 7: Ablation comparison about the V2M task on Human4DiT-Video datset.

| Methods | MPJPE↓ | PA-MPJPE↓ | PVE↓ | Accel↓ |
|---------|--------|-----------|------|--------|
| Only V2M | 53.8356 | 45.9847 | 71.8492 | 6.1748 |
| Uni-Embed. | 50.1287 | 41.3925 | 68.3533 | 5.1539 |
| Our | **43.2689** | **28.1528** | **52.0143** | **4.5647** |

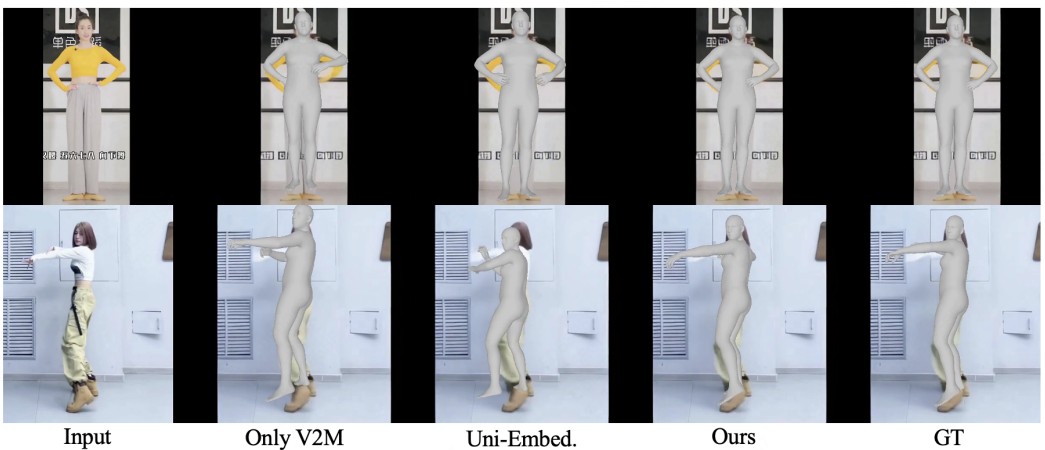

Figure 8: The ablation experiment on the V2M task. We compare our method against single-task training (Only V2M) and the approach without independent embedding (Uni-Embed.).

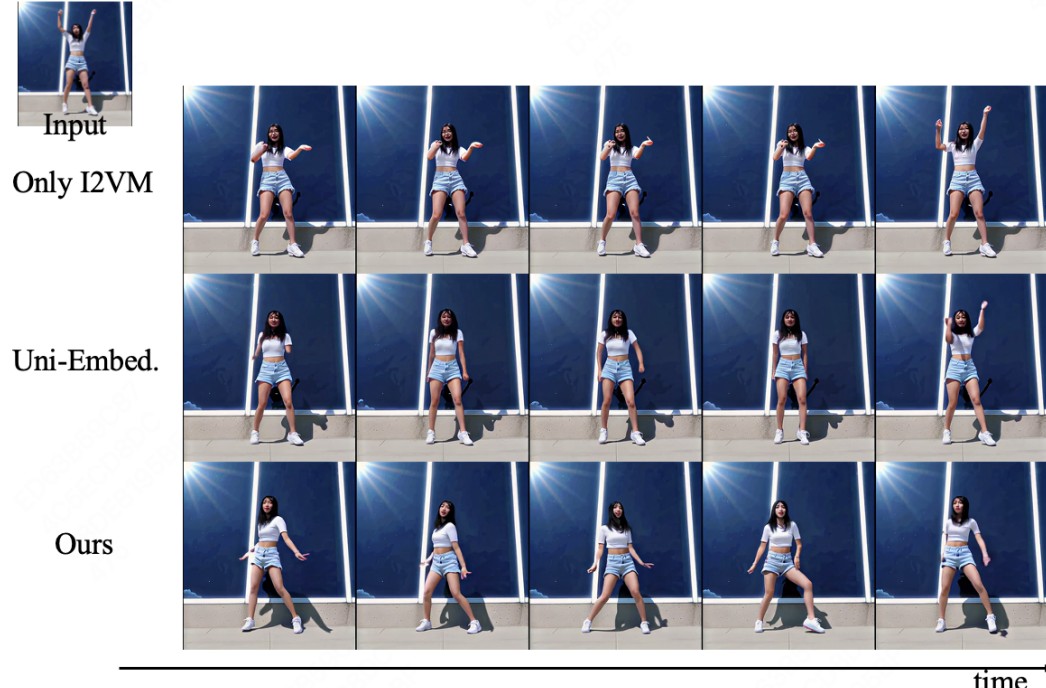

Figure 9: The ablation experiment on the I2VM task. We compare our method against single-task training (Only I2VM) and the approach without independent embedding (Uni-Embed.).

Table 8: Ablation comparison on the I2VM task. For motion-video consistency, GVHMR is used to extract motions as pseudo ground truth. Motion diversity is assessed using the metrics from Duolando. The generated videos are evaluated with VBench, focusing on appearance (App.) and visual motion (Mot.) aspects.

| Method | Video-Motion Consistency | | | | Motion Diversity | | Video Quality | |
|---|---|---|---|---|---|---|---|---|
| | MPJPE ↓ | PA-MPJPE↓ | PVE↓ | Accel↓ | FID↓ | DIV↑ | App.↑ | Mot.↑ |
| Only I2VM | 56.78 | 46.11 | 68.03 | 5.9609 | 51.6599 | 7.3109 | 0.8220 | 0.8577 |
| Uni-Embed. | 53.98 | 38.37 | 59.55 | 4.8837 | 40.7963 | 10.2315 | 0.8387 | 0.8926 |
| Ours | **41.30** | **30.95** | **47.31** | **3.9984** | **27.3984** | **12.2522** | **0.8516** | **0.9441** |

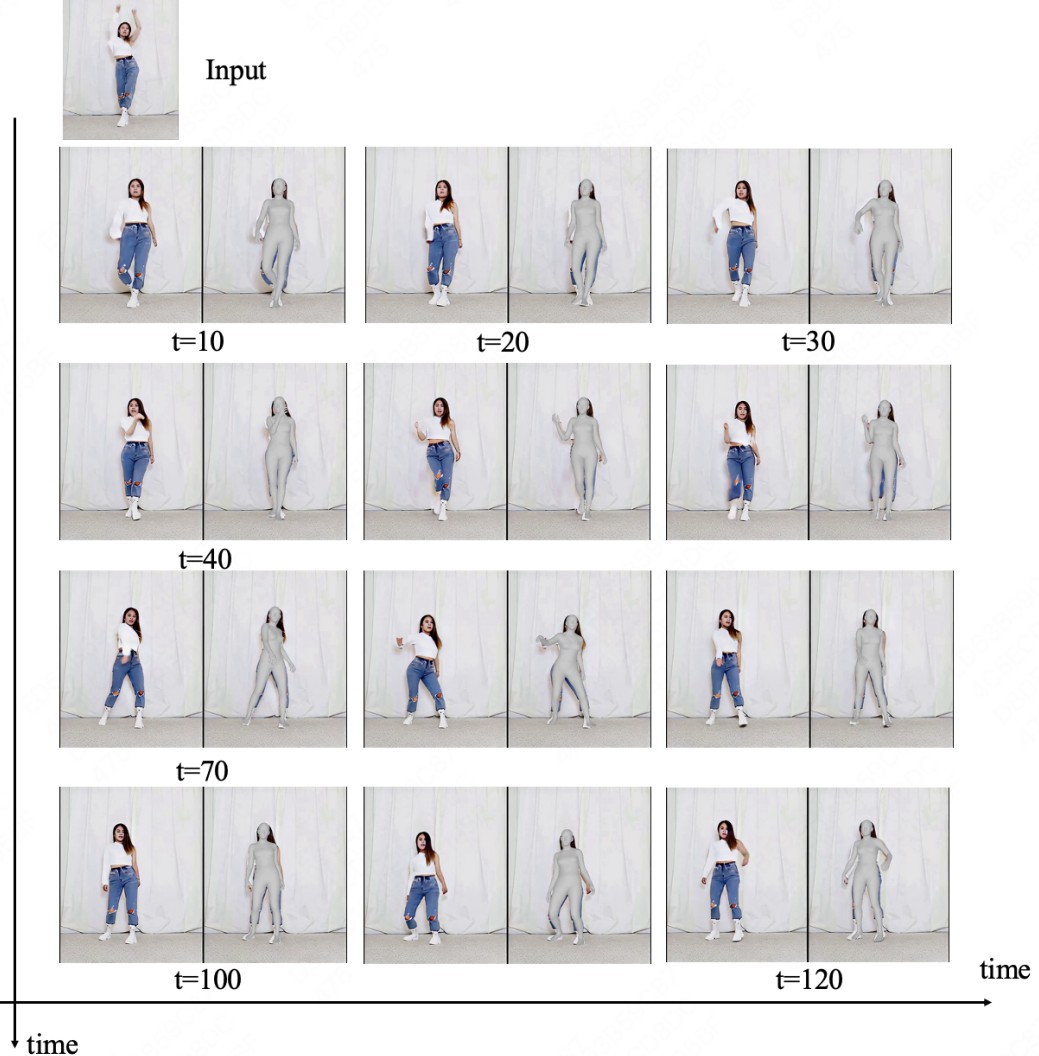

Figure 10: We validate our method at a resolution of 512x512 with 121 frames on I2VM task. The results preliminarily demonstrate our method's potential to handle long sequences.

**Effectiveness of Independent Embedding.** We employ independent embeddings for discrete tokens and positions to mitigate distribution disparities between the two modalities, thereby avoiding confusion when simultaneously outputting video and motion tokens within the same transformer model. The results are shown to achieve superior visual performance in Fig. 8 and Fig. 9. The conclusions drawn from Tab. 8 and Tab. 7 are consistent with the visual results, underscoring the effectiveness of independent embedding.

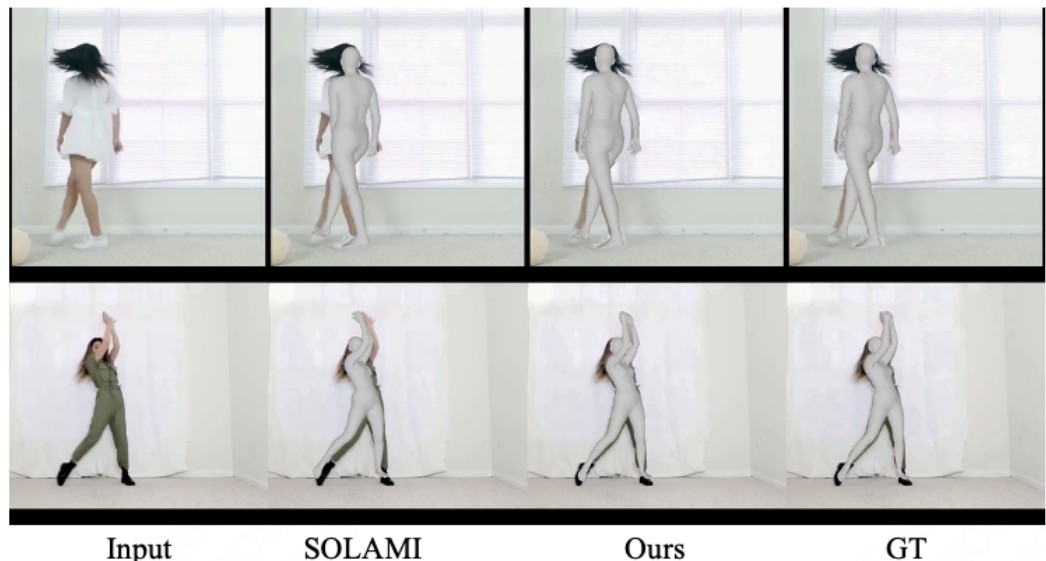

Input          SOLAMI          Ours          GT

Figure 11: Comparison of tokenizers.

Table 9: Comparison of I2VM tasks. Breakdown of VBench metrics on generated videos.

| Method | Appearance | | | | Visual Motion | | | |
|---|---|---|---|---|---|---|---|---|
| | Subj. ↑ | Back. ↑ | Aest.↑ | Overall↑ | Mot.Sm. ↑ | Tem.Fl. ↑ | Dyn.Deg.↑ | Overall↑ |
| Baseline | 0.9535 | 0.9512 | 0.5166 | 0.8071 | **0.9951** | **0.9928** | 0.45 | 0.8126 |
| Ours | **0.9646** | **0.9621** | **0.6281** | **0.8516** | 0.9939 | 0.9884 | **0.85** | **0.9441** |

**Longer Videos.** As a pioneering work in joint modeling 3D motions and 2D videos, our main experiments are conducted at a resolution of 512x512 with 65 frames. To validate the model's potential, we also perform experiments at a resolution of 512x512 with 121 frames. As shown in Fig. 10, our method can handle longer sequences, laying the foundation for extending to more modalities and enabling fine-grained control (see Sec. D).

## C    MORE RESULTS

**3D Motion Tokenizer.** As illustrated in Fig. 11, we present a comparison with SOLAMI. SOLAMI is also a SMPLX-based method that divides the human body into multiple parts, handling translations independently. Our approach achieves more consistent reconstruction results, proving the effectiveness of the temporal expansion strategy.

**I2VM Task.** As shown in Tab. 9, we present the breakdown of VBench metrics for the generated videos. In terms of appearance, our method consistently outperforms the baseline across all metrics. However, in the realm of visual motion, while the baseline performs better in motion smoothness and temporal flickering, the dynamic degree metric reveals that the baseline tends to generate videos with static motion, which accounts for its higher scores in motion smoothness and temporal flickering. Additionally, we present further comparisons with the baseline in Fig. 12, demonstrating that our method can produce more vivid results. From the figure, our approach successfully handles complex motions such as turning around, and provides plausible frontal results.

**V2M task.** We conduct comparisons with 4DHuman and GVHMR on the V2M task. As shown in Tab. 10, our method achieves results comparable to state-of-the-art approaches, even for relatively complex motions, demonstrating the potential of the proposed method. Since we use SMPL-X extracted by GVHMR as pseudo ground truth, GVHMR is excluded from metric calculation.

In addition, we also perform comparisons on classic public datasets, such as 3DPW von Marcard et al. (2018) and RICH Huang et al. (2022) dataset. For fair comparison, we select single-person videos outside the testset as the training set for fine-tuning. Notably, to maintain a unified setting,

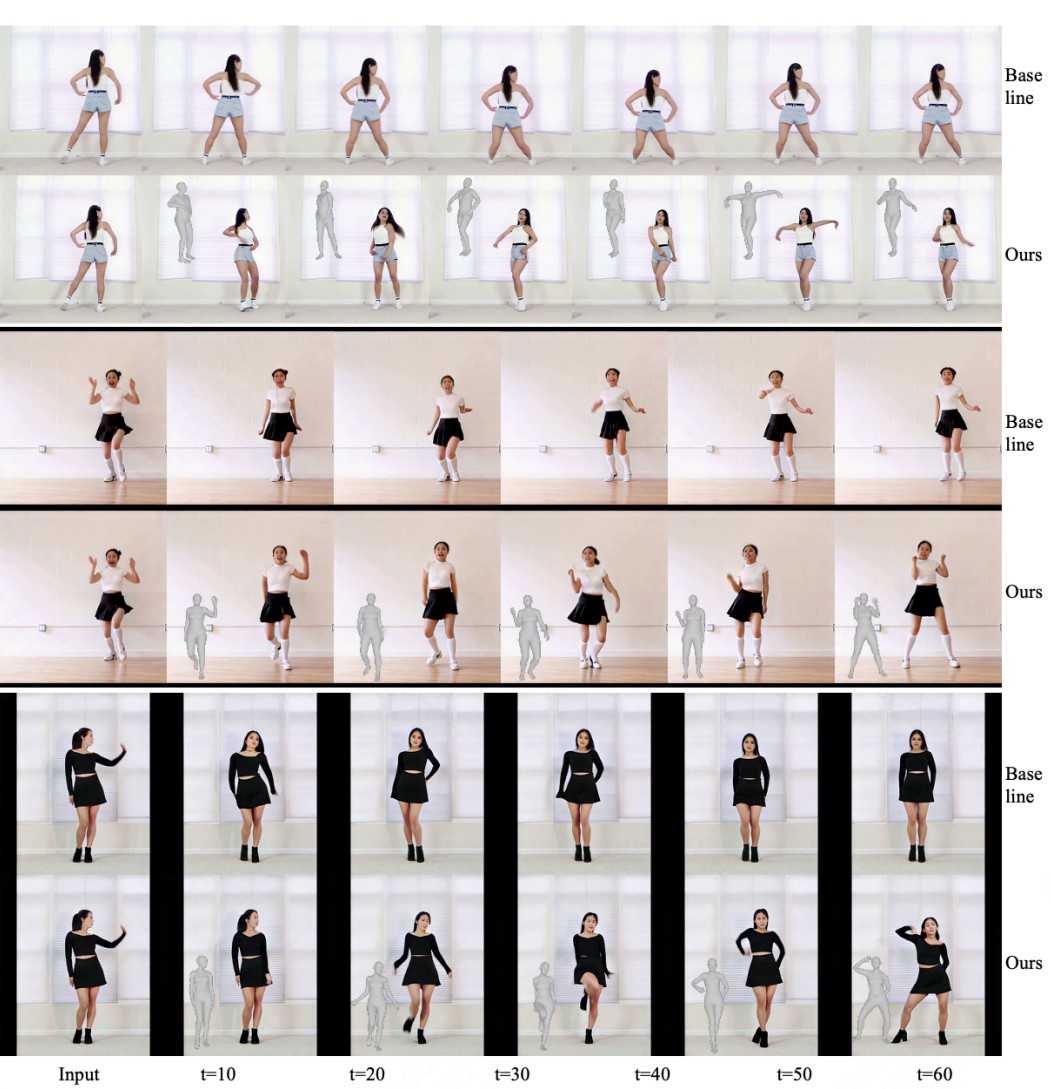

Figure 12: Comparison with the baseline on I2VM tasks. We present results in temporal order from left to right, sampling one frame every 10 generated frames. In our results, the simultaneously generated 3D motions are visualized and presented in the left; the baseline model lacks motion generation capabilities. Given a single image as input, the baseline tends to generate results with minimal motion amplitude, often bordering on stillness.

Table 10: Evaluation of the V2M task on Human4DiT-Video dataset. Since we use SMPL-X extracted by GVHMR as pseudo ground truth, GVHMR is excluded from metric calculation.

| Methods | MPJPE ↓ | PA-MPJPE↓ | PVE↓ | Accel↓ |
|---------|---------|-----------|------|--------|
| GVHMR | - | - | - | - |
| 4DHuman | 56.0701 | 35.4769 | 67.7157 | 10.7393 |
| Our | 43.2689 | 28.1528 | 52.0143 | 4.5647 |

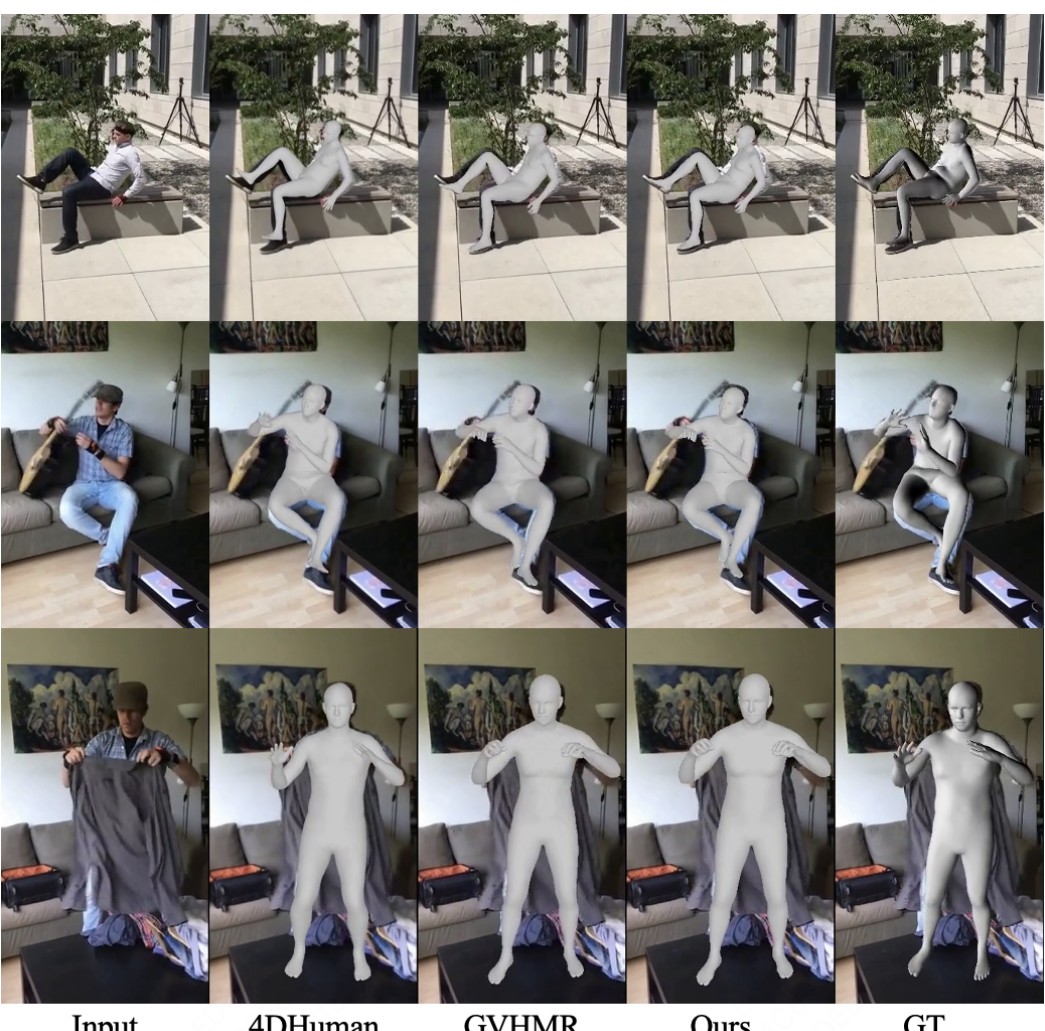

| Input | 4DHuman | GVHMR | Ours | GT |

Figure 13: Comparison of V2M task with different methods on the 3DPW dataset von Marcard et al. (2018). It can be observed that our results are comparable to those of state-of-the-art methods.

the motions within the training set are extracted using GVHMR, but the official ground truth is still used when calculating metrics. Fig. 13 and Fig. 14 reveal that our method achieves results similar to state-of-the-art methods in cases involving occlusion, handheld objects, and lying down poses. Tab. 11 illustrates the same conclusion.

## D    LIMITATIONS AND FUTURE WORK

**Limitations.** This exploratory work marks the initial attempt to jointly model 3D human motions and 2D human videos using an AR model. It currently faces two limitations. Firstly, to leverage large amounts of data available in the wild, we extract 3D motions using GVHMR Shen et al. (2024).

Table 11: Evaluation of the V2M task on 3DPW and RICH dataset. It demonstrates that our approach achieves results comparable to current state-of-the-art methods. (3DPW/RICH)

| Methods | MPJPE↓ | PA-MPJPE↓ | PVE↓ | Accel↓ |
|---|---|---|---|---|
| 4DHuman | 76.33/65.22 | 64.05/55.25 | 82.57/112.37 | 5.458/8.963 |
| GVHMR | **58.97/53.65** | **41.82/46.53** | **73.03/65.36** | **4.190**/4.218 |
| Our | 65.43/58.39 | 50.21/52.62 | 77.14/73.00 | 4.29/**4.182** |

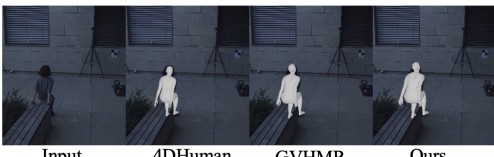 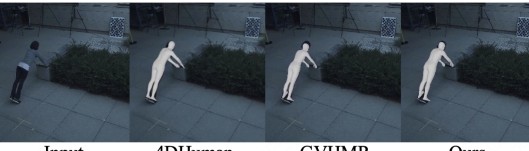

Input    4DHuman    GVHMR    Ours      Input    4DHuman    GVHMR    Ours

Figure 14: Comparison of V2M task with different methods on the RICH dataset Huang et al. (2022). It can be observed that our results are comparable to those of state-of-the-art methods.

However, the accuracy and consistency of these extracted motions are inherently inferior to those captured with specialized hardware in controlled laboratory environments. For instance, MVHumanNet Xiong et al. (2024) utilizes multi-view cameras in laboratory settings for SMPL-X fitting. As a result, our current work does not fully surpass the SOTA methods in all aspects. Nevertheless, existing experiments have already demonstrated the potential of our proposed method, achieving results comparable to existing methods. Secondly, since most large-scale in-the-wild datasets, including Human4DiT-Video, consist only of single-person data, our current work is evaluated using single-person cases.

**Future Work.** Our method pioneers the joint modeling, optimization, and generation of 3D human motions and 2D human videos using an AR model. Existing experiments validate the effectiveness of the proposed approach, with several potential avenues for future expansion.

- **Input Modality:** Currently, our task focuses only on the videos and 3D motions modalities. Leveraging the strengths of large language models (LLMs), future directions could involve integrating additional modalities such as text, audio, and camera parameters. This would enable controlled generation and comprehension of videos that encompass human-object-scene interactions or multi-person dialogues.

- **High-Quality Data:** The entire training process can be divided into multiple stages. For example, given the limited availability of high-quality data, we can initially use large amounts of low-quality datasets to help the model understand the correspondence between videos and motions. Subsequently, high-quality data can be employed for further training to enhance performance. Also, akin to Sapiens Khirodkar et al. (2024), task-specific finetuning can be applied to further enhance the effectiveness of downstream sub-tasks.

- **Better Representation:** Currently, we utilize discrete representations for both videos and 3D motions. Experimental results indicate that discrete representations effectively convey and reconstruct information for relatively simple motions. However, for videos, discrete tokens yield lower visual quality compared to continuous representations such as VAE Wan et al. (2025). Therefore, inspired by Show-o2 Xie et al. (2025), we consider employing continuous flow-based optimization for visual information within a unified multimodal model.

- **Fine-Grained Control:** Our current work does not include specialized handling of faces and hands. In the future, we could segment the human body into head, hands, and body components, using a transformer to unify different driving conditions, thereby enabling more granular modeling and control.

- **Multi-Human Videos:** Current experiments have demonstrated the capability to control and capture human translation and poses. Thus, we can extend beyond single-human videos by processing SMPL-X parameters of multiple people within one motion tokenizer, facilitating multi-human generation and understanding.

**LLM Usage.** LLM tools are used for improving grammar. First, we manually complete the writing of the entire paper, and then use LLM to adjust incorrect words and sentence structures.

## E    VIDEO DEMO

We provide generated results in the demo.

