# OpenReview forum: "UniMo: Unifying 2D Video and 3D Human Motion with an Autoregressive Framework"
_ICLR.cc/2026/Conference — ICLR 2026 Conference Withdrawn Submission_

### Official Review · Reviewer_dd8t · 2025-10-29

**Soundness:** 3
**Presentation:** 2
**Contribution:** 2
**Rating:** 4
**Confidence:** 3

**Summary:**

The paper presents UniMo, a GPT-like model for 2D human video and 3D human motion generation. It unifies Image-to-video-and-motion (I2VM) and video-to-motion (V2M) into one model. To achieve this, it designs a motion VQVAE to quantify the motion sequence into discrete tokens that align with the number of visual tokens. It also adopts several designs to enable GPT to handle tokens of different modalities, including separate embeddings and the use of special tokens to indicate tasks. Experiments show the better performance of both video and motion generation compared to the baseline methods.

**Strengths:**

1. The tasks the paper tries to address are interesting and novel to me. It's intuitive to expect co-improvement by modeling the 3D motions and 2D videos in one framework. The Video2Motion would also be useful to extract 3D motion from in-the-wild videos.
2. The performance reported in the table and shown by the demo video is impressive.
3. The design of a GPT-like model and independent embeddings to unify two modalities is straightforward and intuitive. The experiments demonstrated the effectiveness of the proposed designs.

**Weaknesses:**

1. One major issue is that the paper keeps mentioning "understanding task" several times. But I cannot find any design or experiment related to understanding, which I expect that the model will receive video or motion and text as input, and generate text. The use of LLM is also improper. I would suggest describing the model as a GPT-like model rather than a LLM-based model.
2. The design to extend the temporal dimension to match the number of visual tokens is relatively naive. The author mentioned that using body parts vq will introduce complexity due to the use of different codebooks. But it is not necessarily the case. Partition-based on body part is exactly similar to patchfication. It can use one large code book and can also extend the number of tokens for one frame. The author might conduct related ablation experiments to support this design decision.
3. The use of independent embeddings for different modalities is also a relatively naive approach that is well explored for multi-modal GPT. I wonder what the differences are compared to the related work about multi-modal GPT, like text and image generation [1].

[1] Team, Chameleon. "Chameleon: Mixed-modal early-fusion foundation models." arXiv preprint arXiv:2405.09818 (2024).

**Questions:**

Given the advances of the video generation model, how can the method possibly be built upon those strong video foundation models? How the proposed framework can advance video generation or 3D human motion estimation with the current progress of video models.

---

### Official Review · Reviewer_QiuQ · 2025-10-31

**Soundness:** 3
**Presentation:** 3
**Contribution:** 2
**Rating:** 4
**Confidence:** 4

**Summary:**

This paper proposes joint modeling of 2D human videos and 3D motion.

The proposed framework combines tokens from both modalities into unified tokens and trains an autoregressive model using multiple decoders. Specifically, two tasks (Image-to-Video-and-Motion (I2VM) and Video-to-Motion (V2M)) are defined, each with its own task-specific token arrangement.

In addition, motion tokens are enhanced through temporal expansion with a scaling factor. The positional embedding combines absolute and relative embeddings to preserve the relationships between the two modalities.

Experimental results show that the proposed method generates more dynamic videos and achieves high consistency in the generated motion compared to previous approaches.

**Strengths:**

- By performing joint modeling of 2D human videos and 3D motion, the framework ensures consistency between video and motion. Task-specific token arrangements are defined for each task, enabling effective generation.

- By increasing the resolution of motion tokens using a scaling factor (s), the method improves motion prediction accuracy. Although the theoretical basis is limited, experimental results demonstrate its effectiveness.

- The paper is easy to follow.

**Weaknesses:**

- Possible evaluation bias stemming from reliance on GVHMR pseudo ground truth.

The framework is both trained and evaluated on data derived from GVHMR, which also serves as the reference for metric computation. This design could introduce a subtle circular bias, as the model may partially learn to replicate GVHMR characteristics rather than demonstrating genuine generalization. A clearer discussion on how this dependency is handled or additional evaluation using independent ground-truth would strengthen the validity of the results.


- Unclear fairness of visual-quality evaluation due to post-processing.

The reported VBench results seem to rely on SeedVR post-processing, which can noticeably enhance visual fidelity. As a result, it’s not entirely clear how much of the improvement comes from the proposed model itself. It would be very helpful if the authors could clarify this point or show an additional comparison without post-processing to give a fairer view of the method’s intrinsic performance.


- Insufficient empirical support for the interleaving-token effect.

Equation (2) claims that interleaving visual and motion tokens improves the model’s integration ability. However, there is no controlled comparison between interleaved and non-interleaved configurations to verify this effect. Adding such an ablation, even briefly, would provide stronger evidence for one of the paper’s key design choices.


- Limited diversity of baseline comparisons, especially for the I2VM task.

The experimental section primarily discusses a few baselines, such as SOLAMI and Cosmos. I was wondering if the paper could benefit from a broader set of comparisons with recent video-to-motion or motion-to-video methods. This might help clarify where the proposed joint modeling provides clear advantages and where it may still face challenges. Such an addition could further highlight the method’s strengths and make the overall contribution more compelling.

**Questions:**

- The overall structure appears to be a combination of Duolando and Cosmos AR. Besides the proposed token configuration, the paper’s unique contribution compared to previous works is not clearly described. This should be clarified.

- The ablation study on the 3D motion tokenizer (i.e., ablation on s) should be provided in more detail in the main manuscript. It is recommended to include Appendix Tab. 5 in the main paper. In particular, it suggests that the performance is largely affected by the codebook size than by the scaling factor s. Therefore, additional analysis on this issue is necessary.


- Expanding the token size will increase the model size and computational cost, however, no discussion is provided in the manuscript.

- In Eq. 2, the authors state that the interleaved visual and motion tokens strengthen the model’s integration capability. While the intention is understandable, the authors should provide supporting evidence showing that interleaving indeed improves performance. Currently, there is no such evidence presented.

- During the two-stage learning process, the tokenizer is frozen. Would it not be advantageous to train the entire network, including the tokenizer, during the second stage for potential performance improvements?

- The comparison with existing work is limited to SOLAMI. Although a direct quantitative comparison with other Human Video–Motion Joint Tasks mentioned in the Related Works may be difficult, since the paper uses GVHMR as GT, a qualitative comparison with prior works that also use this GT as input would enrich the analysis and strengthen the paper.




- A more detailed comparison with prior studies would strengthen the paper. Although the purpose is joint modeling, qualitative comparisons with Video-to-Motion or Motion-to-Video generation methods would help clarify how effective the joint modeling is for each task (e.g., whether it outperforms, matches, or shows specific limitations compared to existing approaches).

- The proposed method is limited to single-person cases (as mentioned by the authors). This could be discussed.




- Several (minor) editing errors are found throughout the manuscript:

-- p.5 L238: Please clarify that "cascade M along the last dimension channel" will result in T \times (63+10+3+3) sized feature.

-- p.5 L239: s is not termed properly (e.g., scaling factor).

-- The citation format throughout the paper is incorrect. Please use the appropriate LaTeX commands (e.g., \cite, \citep, \citet, etc.).

-- Many published papers are cited as arXiv preprints (e.g., Duolando was published in ICLR 2024).

-- Fig.1 caption: "joint modeling 2D human" -> "joint modeling of 2D human"

-- p.3: "modal integration" -> "modality integration"?

-- p.5 L262: 8x16x16 -> 8 \times 16 \times 16

-- All equations should end with a comma if they appear mid-sentence or with a period if they appear at the end of a sentence

-- p.7 L340: sg means -> sg(\cdot) means

-- Fig.5 caption: s=1 -> $s$ = 1

---

### Official Review · Reviewer_DDhb · 2025-10-31

**Soundness:** 3
**Presentation:** 3
**Contribution:** 3
**Rating:** 4
**Confidence:** 3

**Summary:**

This paper introduces UniMo, a unified autoregressive model for joint modeling of 2D human videos and 3D human motions. Unlike prior methods that treat video generation and motion capture as separate tasks or rely on 2D motion maps as intermediates, UniMo directly integrates both modalities in a single transformer framework, enhanced with a fine-grained 3D motion tokenizer, independent modality embeddings, and a multi-task joint training strategy. Experiments demonstrate that UniMo achieves competitive or superior results on both video quality and motion reconstruction accuracy.

**Strengths:**

- The motivation is clear and well-grounded, unifying I2VM and V2M provides a natural way to achieve mutually benefits between reconstruction and generation quality, and such joint modeling has been shown to be effective in other domains as well (e.g. GENMO [ICCV 2025]).

- The design of key components (e.g., independent embeddings, SMPL-X token expansion) is well explained and ablated.

- The framework consistently improves both motion reconstruction and generation quality, demonstrating the effectiveness of the proposed joint modeling approach.

**Weaknesses:**

- Clarity and organization could be improved. For example, the meaning of s=1 in Figure 5 could be clarified directly in the caption for easier reference. Several important quantitative results and ablations, such as the I2VM-only and V2M-only settings, are presented only in the supplementary material, though they are essential for substantiating the claimed benefits of joint modeling.
- The Cosmos baseline used for comparison appears relatively weak, as human motion dynamics constitutes only a small fraction of its training distribution, and it is conditioned solely on images without additional context for dancing. This may limit the fairness of comparison, particularly since UniMo is fine-tuned specifically for video generation with human motion.
- It would be valuable to include a comparison against a fine-tuned Cosmos variant that leverages alternative representations such as motion maps or keypoint sequences, to better assess whether explicit 3D motion modeling is truly necessary for improved video generation quality.
- The paper could provide more discussion on computational trade-offs in the AR model when using expanded motion tokens.

**Questions:**

- The expanded SMPL-X motion tokenization improves reconstruction significantly but how much computational overhead does this introduce for the AR model? Are all 36 motion tokens generated autoregressively per frame, and how does this scale with sequence length?


- Could the I2VM module alone be used for V2M inference by replacing video tokens with ground truth as input? If so, have the authors tested its V2M reconstruction capability without the explicit V2M branch?


- In the ablation studies, does the “uni-emb” configuration correspond to a uni-emb + single-task setting, or is it uni-emb+trained jointly?


- The loss curve in Figure 7 doesn’t show a clear advantage of V2M over I2VM during training, when it seems like V2M is a relatively easier task for the model to learn. Could the authors provide some insight into why this is the case?


- How does having I2V + V2M as multi-task training compare to I2VM + V2M?

- Minor: The wording of "LLM-based framework" is confusing, at it seems to imply you are using an LLM backbone. "LLM-style" or "autoregressive" framework might be more accurate.

---

### Official Review · Reviewer_pm92 · 2025-10-31

**Soundness:** 3
**Presentation:** 3
**Contribution:** 3
**Rating:** 6
**Confidence:** 4

**Summary:**

The paper proposes UniMo, an autoregressive model that unifies 2D human video and 3D SMPL-X motion for both generation and analysis. The method tokenizes visual and motion streams into one token sequence while using separate vocabularies and positional embeddings to reduce cross-modal interference. A dedicated VQ-VAE motion tokenizer quantizes SMPL-X parameters and reconstructs them with multi-expert decoders; a temporal-expansion scheme balances token counts with video. After training the tokenizer, it is frozen, and the AR model is jointly trained on image-to-video-and-motion and video-to-motion tasks. Experiments on Human4DiT-Video, 3DPW, and RICH show strong generation quality and reliable motion recovery. Ablation studies confirm the effectiveness of the key components and demonstrate that multi-task training yields mutual benefits across the two tasks.

**Strengths:**

1. The motivation and task are clearly defined, and the proposed approach is technically sound and promising.
2. Extensive comparisons and ablation studies convincingly demonstrate the method’s effectiveness and isolate the impact of key components.

**Weaknesses:**

1. The paper compares the proposed motion tokenizer to SOLAMI, but SOLAMI employs multiple VQ-VAEs for body, hands, and inter-character relative transforms, whereas this work focuses only on body motion. The absence of SOLAMI’s detailed settings further undermines the validity of the comparison and may bias the results.
2. Since the main baselines generate only video, the current I2VM results are not sufficient to support the claim that motion supervision improves video generation. The evaluation lacks strong image-to-video baselines (e.g., Wan-2.1, Hunyuan-Video), weakening the empirical case.
3. The design choice of expert decoders within the motion tokenizer is not empirically justified. There are no ablations comparing against a single shared decoder, different numbers of experts, or alternative routing strategies.

Minor: At line #233, when introducing SMPL-X, it’s suggested to cite the complete SMPL-X definition (including face and hands). Even if this work uses only the body subset, a full citation will reduce potential misunderstanding

**Questions:**

1. I am unclear about the meaning of using two RoPEs for video and motion. Does this imply distinct RoPE configurations (e.g., different frequency bands) or different index designs per modality? Please clarify how temporal alignment between modalities is maintained when separate RoPEs are employed, especially during interleaved autoregressive prediction.
2. At line #254, the paper states that “with a relatively small number of parameters for the motion tokenizer, expanding the tokens only imposes a small burden.” Why is the computational burden small? Even with a lightweight tokenizer, expanding tokens appears to increase the total sequence length processed by the transformer during autoregressive generation, which should raise latency and compute.
3. a) I am interested in how the layout of visual and motion tokens affects performance. The current description suggests a strictly alternating, token-by-token interleaving. Are the token counts for the two modalities always matched under your settings? If not, how are mismatches handled? b) Prior work (e.g., GLM-4-Voice [1]) reports gains from chunk-level mixing rather than per-token interleaving; have you evaluated chunked schedules or other mixing strategies, and how do they compare?

> [1] Zeng, Aohan, et al. "Glm-4-voice: Towards intelligent and human-like end-to-end spoken chatbot." *arXiv preprint arXiv:2412.02612* (2024).

---

### Note · Authors · 2025-11-14

**Comment:**

Dear reviewers,
Thank you for your valuable suggestions.
We will further improve our study.

**Withdrawal Confirmation:**

I have read and agree with the venue's withdrawal policy on behalf of myself and my co-authors.